# Cost-Effective Synthesis of Efficient CoWO_4_/Ni Nanocomposite Electrode Material for Supercapacitor Applications

**DOI:** 10.3390/nano10112195

**Published:** 2020-11-04

**Authors:** Kannadasan Thiagarajan, Dhandapani Balaji, Jagannathan Madhavan, Jayaraman Theerthagiri, Seung Jun Lee, Ki-Young Kwon, Myong Yong Choi

**Affiliations:** 1Solar Energy Lab, Department of Chemistry, Thiruvalluvar University, Vellore 632 115, India; kthiyagarajanmphil6@gmail.com (K.T.); baladgp@gmail.com (D.B.); 2Department of Chemistry and Research Institute of Natural Sciences, Gyeongsang National University, Jinju 52828, Korea; j.theerthagiri@gmail.com (J.T.); venus272@gnu.ac.kr (S.J.L.)

**Keywords:** cobalt tungstate, CWO–Ni composite, supercapacitors, charge–discharge studies, wet chemical method

## Abstract

In the present study, the synthesis of CoWO_4_ (CWO)–Ni nanocomposites was conducted using a wet chemical method. The crystalline phases and morphologies of the Ni nanoparticles, CWO, and CWO–Ni composites were analyzed using X-ray diffraction (XRD), scanning electron microscopy (SEM), transmission electron microscopy (TEM), and energy-dispersive X-ray spectroscopy (EDAX). The electrochemical properties of CWO and CWO–Ni composite electrode materials were assessed by cyclic voltammetry (CV), and galvanostatic charge–discharge (GCD) tests using KOH as a supporting electrolyte. Among the CWO–Ni composites containing different amounts of Ni1, Ni2, and Ni3, CWO–Ni3 exhibited the highest specific capacitance of 271 F g^−1^ at 1 A g^−1^, which was greater than that of bare CWO (128 F g^−1^). Moreover, the CWO–Ni3 composite electrode material displayed excellent reversible cyclic stability and maintained 86.4% of its initial capacitance after 1500 discharge cycles. The results obtained herein demonstrate that the prepared CWO–Ni3 nanocomposite is a promising electrode candidate for supercapacitor applications.

## 1. Introduction

Natural resource depletion and global warming pose a serious threat to both humans and the environment; therefore, developing efficient energy conversion and storage systems is essential [1]. Among different renewable energy sources, supercapacitors have been demonstrated as excellent energy storage devices. Supercapacitors are environmentally friendly and exhibit various attractive properties, including high power density, stability over long cycles, and high charge–discharge (CD) rates [2,3,4,5,6,7,8,9,10]. Supercapacitors are also recognized as electrochemical capacitors. They can be classified into two types based on their energy storage mechanism, i.e., electrical double-layer capacitors (EDLCs) and pseudocapacitors. In the case of EDLCs, the energy is stored between the electrode/electrolyte interfaces on an effective double layer [11]. Owing to their large specific areas and abundant pore-like structures, various C-based materials have been applied as electrodes for EDLCs [12]. In contrast, pseudocapacitors store energy via Faraday redox reactions on the electrode and electrolyte. Metal oxides/hydroxides [13] and conducting polymers [14] are widely used as electrodes for redox supercapacitor applications.

Because of their multiple oxidation states, excellent conductivity, and high specific capacitance (SC), transition metal oxides play important roles in electrochemical processes. Oxides of Ru [15], Fe [16], Co [17], Ni [18], and V [19], among others, have been employed as electrodes for pseudocapacitors. Additionally, mixed composites of transition metal tungstates have attracted considerable attention for applications in electrochemistry, catalysis, pigments, luminescent materials, and sensors. Ternary metal oxides containing two different metal cations have also been proposed as potential electrode materials for supercapacitors. The combination of two dissimilar metal cations is believed to significantly improve the conductivity of ternary metal oxides. Moreover, the existence of multiple oxidation states improves the electrochemical properties of these materials [20]. Hence, we investigated the potential supercapacitor applications of CoWO_4_ (CWO) in the present study.

In recent years, various metal tungstate oxides, including CoWO_4_ [21], reduced graphene oxide (rGO)/CoWO_4_ [22], Co_3_O_4_@CoWO_4_/reduced graphene oxide (rGO) [23], three-dimensional nanoporous ZnWO_4_ [24,25], NiCo_2_O_4_@NiWO_4_ [26], FeWO_4_ [27], and NiWO_4_/rGO [28], have been used in supercapacitors. Notably, Co and Ni-based electrode materials are particularly well known for such applications due to their superior electrochemical activity [29,30]. For instance, Sarma et al. reported a simple electrodeposition technique, which was used to fabricate an NiO/Co_2_O_3_ composite electrode exhibiting an SC value of >400 F g^−1^ at 20 mV s^−1^. The conducted stability test involving continuous cyclic voltammetry (CV) revealed that the synthesized material retained >50% capacitance after 200 CD cycles [31]. Furthermore, Lu et al. [32] prepared porous nickel–cobalt oxide nanosheets employing a simple electrochemical method. The material was used as an electrode, which displayed remarkable specific capacitance of 453 F g^−1^ at 5 mV s^−1^ and 506 F g^−1^ at 1 A g^−1^. Lai et al. [33] synthesized nickel–cobalt hydroxide nanoarrays/carbon nanofibers, exhibiting a maximum specific capacitance value of 1378 F g^−1^ for nanorod arrays and 1195 F g^−1^ for nanosheet arrays at 1 A g^−1^.

Metal tungstates have multiple oxidation states, improving their electrochemical performance in energy storage devices [20]. In this work, CoWO_4_ and its composites were applied as electrode materials for supercapacitors [21,22,23]. We speculated that CoWO_4_-based electrode materials could be used for electrolysis of water and for generation of green renewable energy storage systems. Hence, we fabricated a composite of CoWO_4_ and Ni nanoparticles using a wet chemical method. Pleasingly, an improvement in the electrochemical properties and cyclic stability was observed upon the addition of Ni nanoparticles to CoWO_4_, demonstrating the potential of the composite for supercapacitor applications.

## 2. Materials and Methods

### 2.1. Materials

Nickel chloride (NiCl_2_·6H_2_O), cobalt chloride (CoCl_2_·6H_2_O), sodium tungstate (Na_2_WO_4_·2H_2_O), sodium hydroxide pellets (NaOH), ethylene glycol (EG, C_2_H_6_O_2_), and potassium hydroxide pellets were purchased from SDFCL, India. Hydrazine hydrate (N_2_H_4_), ethanol and polyvinylidene difluoride (PVDF) were obtained from Sigma-Aldrich, USA. *N*-methyl-2-pyrrolidone (NMP) was acquired from HiMedia, India.

### 2.2. Preparation of Ni Nanoparticles

The Ni nanoparticles were fabricated using the method that was previously reported by Wu et al. [34]. Briefly, 1.185 g of NiCl_2_·6H_2_O was dispersed in 40 mL of EG and in 5 mL of N_2_H_4_ in a 100-mL beaker. The solution was stirred for several minutes at ambient temperature. Subsequently, 5 mL of a 1 M NaOH solution was introduced to the reaction mixture. The mixture was stirred at 60 °C for 2 h. The precipitate was then filtered and washed with double-distilled (DD) water and ethanol. The obtained product was dried at 50 °C for 12 h.

### 2.3. Preparation of CoWO_4_/Ni Nanoparticles

Approximately 0.475 g of CoCl_2_·6H_2_O and 0.659 g of Na_2_WO_4_·2H_2_O were dispersed in a 100-mL beaker containing 40 mL of DD water. The resulting solution was stirred for 30 min at ambient temperature. In another 50-mL beaker, 20 mg of Ni nanoparticles was dispersed in 10 mL of ethanol, followed by sonication for 1 h. The Ni nanoparticle solution was subsequently slowly poured into the first solution, and the reaction mixture was stirred for 3 h at ambient temperature. The resulting precipitate was filtered and washed with DD water and ethanol. The crude material was dried at 80 °C in a hot oven. Pure CoWO_4_/Ni20 was obtained as a light bluish black powder. A similar procedure was followed for the preparation of CoWO_4_/Ni40 and CoWO_4_/Ni80, where 40 and 80 indicate the amount of Ni nanoparticles added into the reaction (mg). Bare CoWO_4_ was synthesized without the addition of the Ni nanoparticles. The molar ratio of Co:Ni in CoWO_4_/Ni20, CoWO_4_/Ni40, and CoWO_4_/Ni80 was approximately 1:0.18, 1:0.36, and 1:0.71, respectively. Hereafter, CoWO_4_, CoWO_4_/Ni20, CoWO_4_/Ni40, and CoWO_4_/Ni80 are denoted as CWO, CWO–Ni1, CWO–Ni2, and CWO–Ni3, correspondingly.

### 2.4. Preparation of CoWO_4_ and CoWO_4_/Ni Nanoparticles as Working Electrodes

The CWO and CWO–Ni nanoparticle composite electrodes that contain varying amounts of Ni were prepared by mixing 80% of the fabricated active material, 10% of carbon black (Super-P), and 10% of PVDF as the binder. All ingredients were mixed with NMP to obtain a paste. The paste was then uniformly loaded on an Ni foam (surface area of 0.5 cm^2^) and was dried overnight. The amount of CWO and CWO–Ni composite nanoparticles loaded on each electrode was 1 mg.

### 2.5. Material Characterization

X-ray diffraction (XRD) analysis was conducted using the Bruker D8 Advance X-ray diffractometer with CuKα radiation (λ = 0.1542 nm) at 4°/min. Fourier transform infrared (FT-IR) measurements were performed using JASCO, FT/IR-4600 (JASCO Int. Co., Ltd., Tokyo, Japan). Scanning electron microscopy (SEM) images were obtained using JEOL-JSM 7600F (JEOL Inc, Peabody, MA, USA). Transmission electron microscopy (TEM) images were acquired using JEOL-JEM 2100F (JEOL, Ltd., Tokyo, Japan). The chemical compositions of the fabricated nanocomposite materials were examined by inductively coupled plasma atomic emission spectrometry (ICP-AES) (ICAP 6000, Thermo Fisher Scientific, Waltham, MA, USA).

### 2.6. Electrochemical Measurements

The electrochemical characterization of the CWO, CWO–Ni1, CWO–Ni2, and CWO–Ni3 nanoparticle composites was performed by CV, galvanostatic charge–discharge tests (GCD), and electrochemical impedance spectroscopy (EIS). All investigations were conducted using an electrochemical system (CHI608E, CH Instruments, Inc., Austin, TX, USA) composed of a standard three-electrode assembly with saturated Ag/AgCl as the reference electrode, a Pt wire as the counter electrode, and CWO or CWO–Ni composites loaded on an Ni foam as the working electrodes. An alkaline solution of KOH (6 M) was employed as the electrolyte solution. The CV analyses were performed in an applied bias window of −0.1–0.5 V, while the GCD evaluation was conducted with the applied bias varying from 0 to 0.42 V. The SC was evaluated using CV via Equation (1) according to the previously described approach [7,35].
(1)$$SC=\frac{Idv}{2mv\Delta V}$$
where *SC* is the specific capacitance in F g^−1^, *Idv* is the integrated area, *m* is the mass (weight) of the electrode material in grams, *v* is the scan rate in mV s^−1^, and *∆V* is the potential window in V.

The SC values of the fabricated electrodes were determined based on GCD using the following Equation (2) [7,36]:(2)$$SC=\frac{I\Delta t}{m\Delta V}$$
where *I (A)* and *t (s)* indicate the discharging current and time, respectively, *m (g)* refers to the amount of loaded nanocomposite nanoparticles, and Δ*V* denotes the applied bias difference (V).

## 3. Results and Discussion

### 3.1. XRD Analysis

Figure 1a–c illustrate the XRD patterns of the Ni nanoparticles, CWO, and different composites of CWO–Ni1, CWO–Ni2, and CWO–Ni3. The spectra of the Ni nanoparticles and CWO exhibit distinctive peaks of both Ni and CWO, which are consistent with the standard diffraction patterns (Ni cubic phase, JCPDS # 04-0850; CoWO_4_ monoclinic phase, JCPDS # 15-0867). Furthermore, the peak corresponding to CWO exhibited low intensity, which indicates an amorphous structure and poor crystallization of the CWO material synthesized via the chemical precipitation method. Typically, compared to a crystalline compound, an active material displaying a broad peak as a consequence of an amorphous structure exhibits additional ion transfer [37]. Hence, the prepared CWO–Ni nanoparticle composites with poor crystallinity were expected to show enhanced electrochemical performance.

### 3.2. FT-IR Analysis

The structures of CWO and CWO–Ni3 were confirmed using FT-IR analysis (Figure 2). The strong bands in the low-frequency region (400–1000 cm^−1^) were attributed to the distinctive Co–O, W–O, and W–O–W bridges. Moreover, the IR region below 500 cm^−1^ corresponded to the deformation modes of the W–O bonds in the WO_6_ octahedron or to the deformation of the W–O–W bridges [38]. The appearance of two key bands at 837 and 664 cm^−1^ was ascribed to the O–W–O vibration mode and to the W–O bond stretching frequency, respectively [39]. The absorbance bands at 3429 and 1630 cm^−1^ were attributed to the –OH group of water, which was present in the samples. The band at 604 cm^−1^ corresponded to the stretching vibration of the Ni–O bond, and its presence confirmed the formation of CWO–Ni3.

### 3.3. Surface Morphology Analysis

Figure 3a–c illustrate the SEM images of the Ni nanoparticles, CWO, and CWO–Ni3, respectively. Figure 3a shows a micrograph of irregular spherical Ni nanoparticles exhibiting an evidently incorrect arrangement of layers. Moreover, Figure 3b demonstrates a nanoporous structure of bare CWO, which appears powdery and displays aggregated, clumpy particles. In contrast, Figure 3c,d show the SEM images of the CWO–Ni3 composite, exhibiting an irregular spherical shape and a nanoporous structure. Hence, the SEM analysis results confirmed the formation of the desired composite.

The elemental compositions of the Ni nanoparticles, bare CWO, and CWO–Ni were characterized using energy-dispersive X-ray spectroscopy (EDAX) (Electronic Supplementary Information (ESI), Appendix A). It was found that the Ni nanoparticles exclusively contained Ni and O atoms, where the latter originated from NiO. Furthermore, the EDAX spectrum of bare CWO displayed peaks corresponding to Co, W, and O atoms. No additional signals were detected. Similarly, the spectrum of the CWO–Ni3 composite illustrated in Appendix A verified the existence of Co, Ni, W, and O atoms, with no peaks indicating the presence of other species. These outcomes confirmed the purity of the prepared composites. The corresponding atomic (%) ratios of the elements identified by EDAX (inset of Appendix A) further demonstrated the purity of the synthesized composites. Hence, the XRD and EDAX evaluation confirmed the formation of the Ni nanoparticles, bare CWO, and CWO–Ni3 composite. The inductively coupled plasma mass spectrometry (ICPMS) analysis showed that the molar ratio of Co:Ni in CWO–Ni1, CWO–Ni2, and CWO–Ni3 was 1:0.15, 1:0.34, and 1:0.6, respectively.

### 3.4. TEM Analysis

Figure 4 shows the TEM images of bare CWO and the CWO–Ni3 composite. As it can be seen in Figure 4a, bare CWO exhibited a rocklike structure in the nanometer range. The TEM images of the CWO–Ni3 composite clearly revealed the irregular shape of the material, with the nanocrystal size of approximately 40–50 nm (Figure 4b). Ni displays a spherical shape and is well bound within the nanocrystal structure.

### 3.5. Analysis of the Electrochemical Properties

The electrochemical properties of the electrode materials were investigated using CV, GCD, and EIS. The CV plots of CWO, CWO–Ni1, CWO–Ni2, and CWO–Ni3 were obtained at 10 mV s^−1^ (Figure 5a). Figure 5b demonstrates the CV curves of CWO–Ni3 at scan rates of 10–50 mV s^−1^. Based on the CV curves, the SC values of pure Ni, CWO and the CWO–Ni1, CWO–Ni2, and CWO–Ni3 composites at 10 mV s^−1^ were established at 80, 110, 132, 225, and 251 F g^−1^, respectively. The CV curves of CWO–Ni3 exhibited a strong redox peak, indicating that the SC values were predominantly controlled by the Faradaic redox reactions [40]. Figure 5c shows the SC values of bare CWO and the CWO–Ni1, CWO–Ni2, and CWO–Ni3 composites. It can be seen that the SC increased with increasing Ni amount. Among the investigated CWO–Ni composites, CWO–Ni3 displayed the highest SC value; therefore, it could be applied as an electrode material for supercapacitor applications.

Figure 6a shows the galvanostatic discharge curves of bare CWO, CWO–Ni1, CWO–Ni2, and CWO–Ni3 at a current density of 1 A g^−1^. The discharge curves of all electrode materials increased with increasing amount of Ni. According to the discharge time, the SC values of CWO, CWO–Ni1, CWO–Ni2, and CWO-Ni3 at 1 A g^−1^ were calculated at 128, 145, 195, and 271 F g^−1^, respectively. Furthermore, Figure 6b demonstrates the CD plots of the CWO–Ni3 composite at different current densities (1–9 mV s^−1^). Unlike the discharge curves, the CD plots showed a decrease in the SC values with increasing current density, which was attributed to the decrease in the flow of charged ions into the inner active sites and a reduced rate of the Faradaic redox reaction [41]. Figure 6c demonstrates the reversible capacitance values of bare CWO, CWO–Ni1, CWO–Ni2, and CWO–Ni3. Notably, the capacitance values increased with the increasing amount of Ni that was added to bare CoWO_4_. These outcomes indicated that among the studied composites, CWO–Ni3 exhibited the highest discharge time, making it a suitable candidate electrode material for supercapacitors.

We subsequently evaluated the cyclic stability of the CWO–Ni3 composite with respect to the SC and coulombic efficiency (Figure 7). The CWO–Ni3 active electrode material retained 86.4% of its initial capacitance after 1500 continuous cycles at 5 A g^−1^. Figure 7 shows the calculated coulombic efficiency and SC values of the CWO–Ni3 composite. The initial SC of CWO–Ni3 (242 F g^−1^) at 5 A g^−1^ increased to 252 F g^−1^ up to 400 reversible cycles, corresponding to a 104% rise in SC. The long-term cyclic data demonstrated in Figure 7 indicate that the increase in the SC values for the CWO–Ni3 composite was caused by the entire activation process of the electrode material [42,43] in the electrochemical redox reaction. In the redox process, the electrolyte progressively permeated into the inner segment of the electrode material, increasing the amount of stimulated active sites, and consequently, the SC of the electrode [42]. It can be noticed that after 400 cycles, the SC was slightly decreasing with the increase in cycles due a slight change in resistance of the electrode upon prolonged repeated charge/discharge cycles. The calculated coulombic efficiency of CWO–Ni3 was 100% and was slightly lower in the earlier cycles. The cyclic stability of CWO–Ni3 was evident from the high SC values obtained in this study. Hence, it was further confirmed that the prepared CWO–Ni3 material would be a suitable active electrode in redox supercapacitors.

Furthermore, Figure 8 demonstrates the linear dependence of the CV anodic peak current on the square root of the investigated scan rates. The straight line shown in Figure 8 was plotted according to i α√*v*, indicating the occurrence of a bulk diffusion process via an electrochemical reaction. It is noteworthy that the CV peak current followed the linear Randles–Sevcik equation [44,45,46]. Moreover, an increase in the scan rates resulted in an increase in the CV anodic peak current (Figure 8). The diffusion coefficients of CWO and CWO–Ni3 were estimated from the slope of the Ip vs. √*v* curve using an equation reported in our previous study [41]. The D values for CWO and CWO–Ni3 were determined at 4.8339 × 10^−9^ and 1.7484 × 10^−8^ cm^2^ s^−1^, respectively. Hence, the CWO–Ni3 composite showed a significantly higher D value than CWO.

## 4. Conclusions

CWO–Ni nanoparticle electrode materials exhibiting different wt% of Ni were synthesized using a wet chemical method. The phase formation of the synthesized samples was confirmed by XRD and FT-IR spectroscopy analyses. The SEM surface morphologies of CWO–Ni3 confirmed the co-existence of irregular spherical shaped Ni nanoparticles and nanoporous CWO in its structure. The prepared CWO–Ni3 nanocrystals were also investigated by TEM. Moreover, electrochemical tests, including CV, CD, and EIS, were performed on CWO–Ni3. According to the obtained CD profile, the CWO–Ni3 composite displayed a maximum SC value of 271 F g^−1^ at 1 A g^−1^, which was higher than that of bare CWO (128 F g^−1^). Notably, in a long-term cyclic stability test conducted by applying up to 1500 continuous reversible CD cycles, the CWO–Ni3 composite retained approximately 86.4% of its initial capacitance. Thus, the outcomes of our study clearly indicated that the CWO–Ni electrode material could be applied in redox supercapacitors.

## Figures and Tables

**Figure 1 nanomaterials-10-02195-f001:**
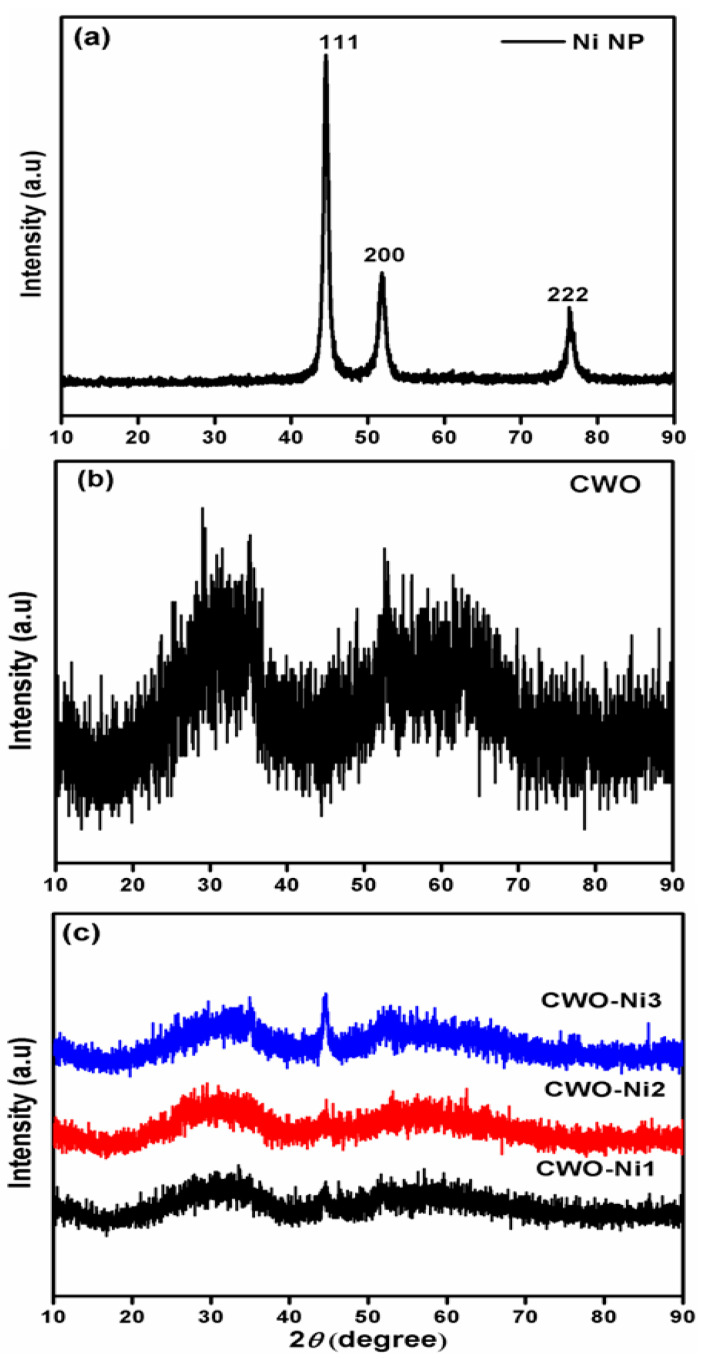
X-ray diffraction (XRD) patterns of the (**a**) Ni nanoparticles; (**b**) CoWO_4_ (CWO); and (**c**) different composites of CWO–Ni1, CWO–Ni2, and CWO–Ni3.

**Figure 2 nanomaterials-10-02195-f002:**
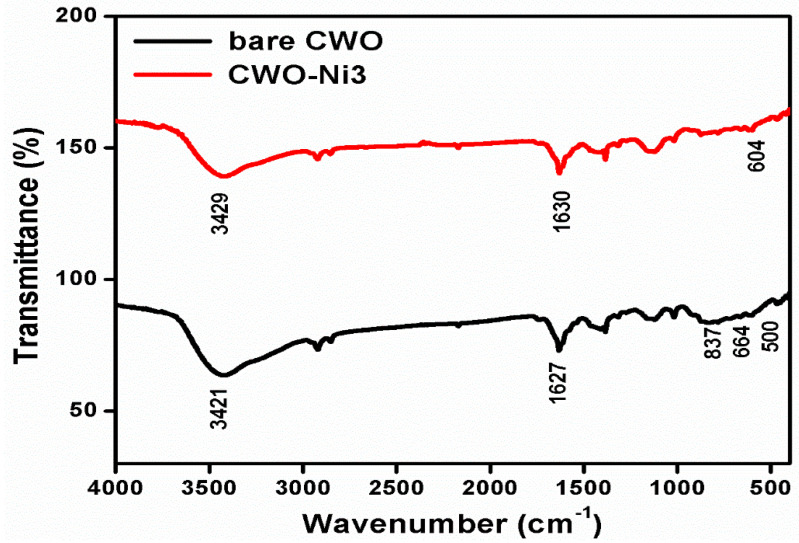
Fourier transform infrared (FT-IR) spectra of bare CWO and CWO–Ni3.

**Figure 3 nanomaterials-10-02195-f003:**
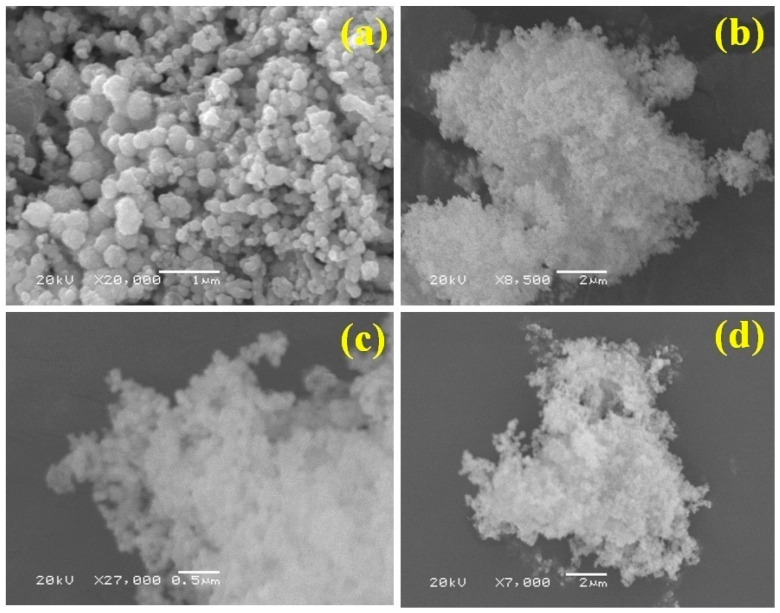
Scanning electron microscopy (SEM) micrographs of the (**a**) Ni nanoparticles, (**b**) bare CWO, and (**c**,**d**) CWO–Ni3 composite.

**Figure 4 nanomaterials-10-02195-f004:**
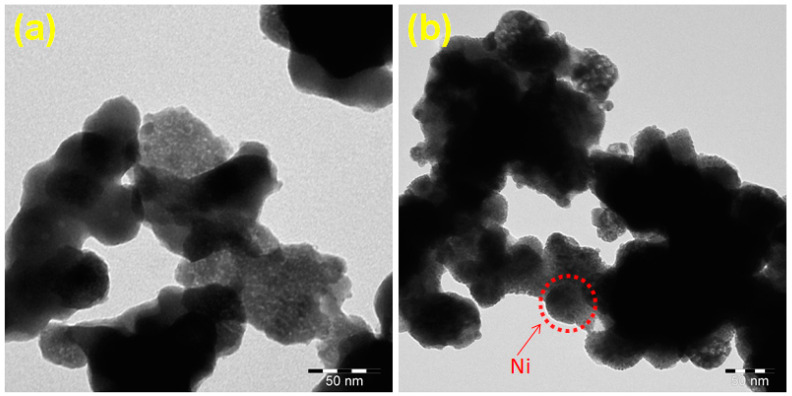
Transmission electron microscopy (TEM) images of (**a**) bare CWO and (**b**) CWO–Ni3.

**Figure 5 nanomaterials-10-02195-f005:**
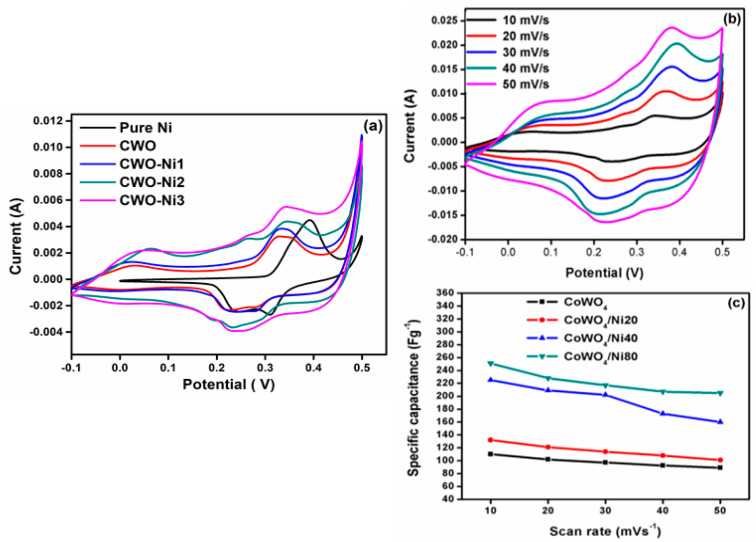
(**a**) Comparison of the cyclic voltammetry (CV) plots of pure Ni, CWO, CWO–Ni1, CWO–Ni2, and CWO–Ni3; (**b**) CV curves of CWO–Ni3 at scan rates ranging from 10 to 50 mV s^−1^; (**c**) comparison of the specific capacitance (SC) values of CWO, CWO–Ni1, CWO–Ni2, and CWO–Ni3 at different scan rates.

**Figure 6 nanomaterials-10-02195-f006:**
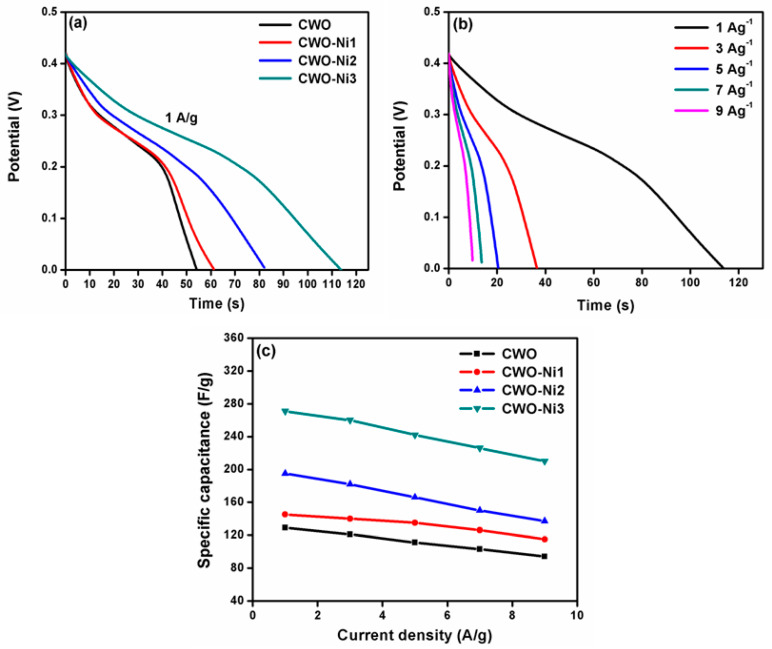
(**a**) Comparison of the charge–discharge (CD) curves of bare CWO, CWO–Ni1, CWO–Ni2, and CWO–Ni3; (**b**) CD plots of CWO–Ni3 at current densities varying from 1 to 9 A g^−1^; (**c**) comparison of the SC values of CWO, CWO–ONi1, CWO–Ni2, and CWO–Ni3 at different scan current densities.

**Figure 7 nanomaterials-10-02195-f007:**
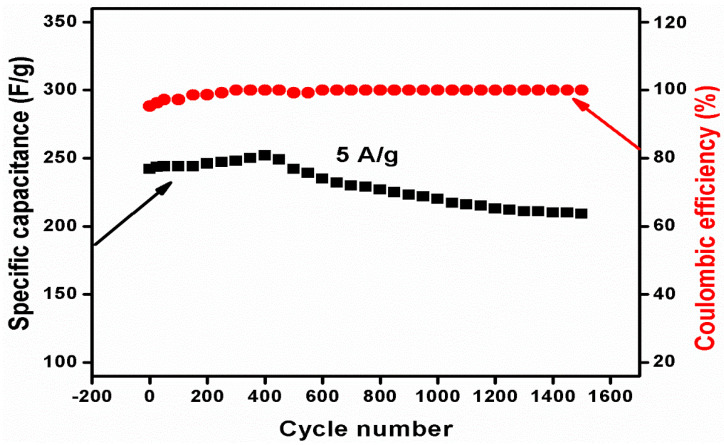
SC and coulombic efficiency vs. cycle number for the CWO–Ni3 electrode material for supercapacitor applications.

**Figure 8 nanomaterials-10-02195-f008:**
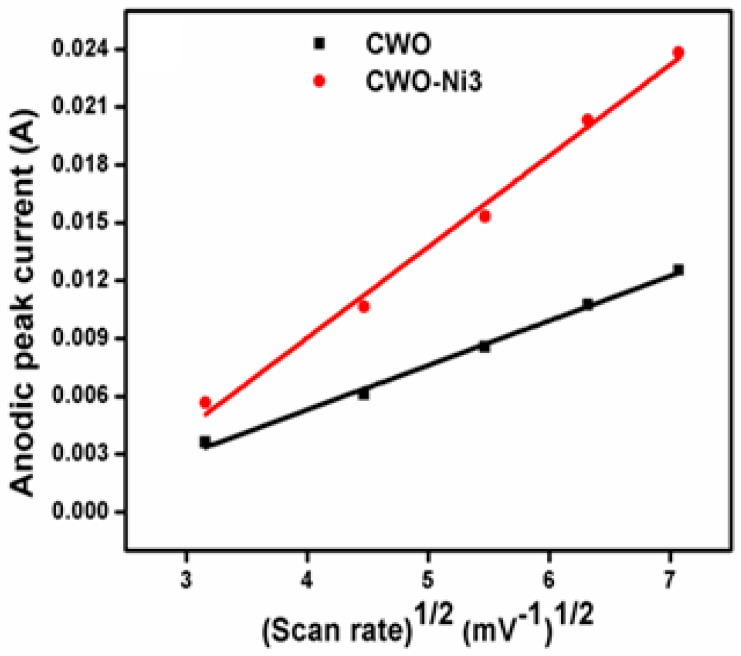
Linear dependence of the CV anodic peak current on the square root of the various scan rates for bare CWO and CWO–Ni3.

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
