# Peer review of "Cost-Effective Synthesis of Efficient CoWO4/Ni Nanocomposite Electrode Material for Supercapacitor Applications"

_nanomaterials, 2020, doi:10.3390/nano10112195_

Round 1
Reviewer 1 Report
The authors are interested in developing supercapacitor material and present results of mixing different amounts of synthesized Ni particles with synthesized CoWO4 material. The authors indicate that supercapacitor materials have high capacitance as a result of double layers on high surface area material and the ability to perform reversible redox reactions with components of the electrolyte. They show results of constant current discharge data in figure 6 to measure the capacitance of the the cell through the relationship that capacitance is equal to the current times the the discharge time divided by the voltage range of the discharge. This data shows that the material with the most Ni included leads to the highest capacitance. In figure 5, they show the results of CVs performed on the material and plot the capacitance of the materials for different CV rates. It is not clear how they did this using the equation SC = I/mass/(CV rate) because the current is varying through the entire CV unless they used the average current, which, if true, should have been tabulated. As an alternative, they could have integrated the area under the CV to get the capacity and then divided by the voltage of the sweep which was 0.6 V.
The authors measure the capacitance of the CoWO4 without Ni, but not the capacitance of the Ni alone. It would have been nice if they plotted the capacitance at a low rate versus Ni content to see if the relationship was linear and to tease out the fraction of the capacitance due to CoWo4 and tht attributed to the Ni material. It was also not clear whether the final form of the Ni particles was pure Ni or some fraction of nickel oxide. The redox reaction of the material with the electrolyte was not provided.
The authors show cycle life data of the material for 1500 cycles, the first 4oo cycles of which the capacity gently rises. After which, the capacity falls at a decelerating rate. No explanation is provided.
The authors take the time to estimate the energy and power per unit volume but do not give details as to how they calculate the volume of the material and it is not clear as to the voltage they used. I must assume they used the density of the materials and specific capacitance they calculated from the other graphs and assumed that it was a symmetric cell and used half of the voltage, but these details again were not provided.
Reviewer 2 Report
The manuscript reports on the preparation of bare CWO and CWO-Ni composites, their characterization by several techniques (XRD, SEM, TEM,…) and their electrochemical study as active electrode materials for supercapacitors. Although the manuscript fits the scope of Nanomaterials, the part dealing with the electrochemical study misleads the concepts of pseudocapacitor electrode and battery electrode. So, I think this part should be revised and amended before publication. I regret, I cannot recommend publication of the manuscript in its current form.
Comments:
1.-The narrow peaks observed in the CVs (Figs. 5a and 5b) and the plateaus observed at ca. 0.25 V on discharge (Figs. 6a and 6b) are characteristics of battery electrode; they cannot be ascribed to pseudocapacitor electrodes. Authors can check the paper by T. Brousse et al. in J. Electrochem. Soc. 162(5) (2015) A5185. The linear dependences found by the Authors for the current on the square root of the potential scan rate for both, CWO and CWO-Ni3 (Fig. 8b) confirm diffusion-controlled kinetics, which is characteristic of battery electrodes. A pseudocapacitor electrode shows linear dependence of the current on the potential scan rate, i.e. surface-controlled kinetics. Authors can check the articles by V. Augustyn et al. Energy Environ. Sci. 7 (2014) 1597 and by I. Aldama et al. J. Electrochem Soc. 165(16) (2018) A4034, and references therein, for understanding the kinetics differences of battery electrodes and pseudocapacitor ones.
2.-In accordance with my first comment, the electrochemical results of CWO and CWO-Ni3 must be discussed in terms of specific charge or specific capacity (C/g or mAh/g); they cannot be discussed in terms of specific capacitance (F/g). The energy density and power density are usually reported for two-electrode cells, not for three-electrode cells. However, if they are maintained in the revised manuscript for comparison of the two types of samples, bare CWO and CWO-Ni composite, they should be recalculated according to the fact that the two samples are battery electrodes and the energy equation should be amended.
3.-I think the specific capacity of the bare Ni particles should be measured. It will allow the discussion of the specific capacity of the composites on the basis of the rule of mixtures, i.e. by comparing the experimental specific capacity of the CWO-Ni3 composite with the specific capacity of each component (CWO and Ni) and the relative content of the two components in the composite.
4.-I suggest deleting the part dealing with the EIS results and discussion. I think this part could mislead the readers. If maintained in the revised manuscript, it should be rewritten according to the new interpretation of the samples as battery electrodes.
5.-Minor corrections:
(i) Scales lack on the intensity axis of Figures 1a and 1c.
(ii) Add any mark in Figure 4 for identifying the two phases, CWO and Ni.
